# Diversity of Freshwater Calanoid Copepods (Crustacea: Copepoda: Calanoida) in Southern Vietnam with an Updated Checklist for the Country

Phuttaphannee Boonmak [1] and Laorsri Sanoamuang [2,3,*]

1. Department of Science and Technology, Faculty of Liberal Arts and Science, Roi Et Rajabhat University, Roi Et 45120, Thailand; phuttaphannee@gmail.com
2. Laboratory of Biodiversity and Environmental Management, International College, Khon Kaen University, Khon Kaen 40002, Thailand
3. Applied Taxonomic Research Center, Faculty of Science, Khon Kaen University, Khon Kaen 40002, Thailand
* Correspondence: la_orsri@kku.ac.th

**Abstract:** The diversity of freshwater calanoid copepods from different habitats in Vietnam is investigated based on our findings from a field expedition in 2012–2013 and literature reviews. We collected 160 samples from 87 sites, including lakes, ponds, roadside canals, rivers, and rice fields in eight provinces of southern Vietnam. A total of 13 species belonging to eight genera and three families were recorded. Among these, four were recorded for the first time in Vietnam (*Mongolodiaptomus malaindosinensis*, *Mongolodiaptomus mekongensis*, *Vietodiaptomus blachei*, and *Pseudodiaptomus siamensis*). One unidentified taxon (*Tropodiaptomus* sp.) probably belongs to an undescribed species. Both *Eodiaptomus draconisignivomi* and *M. malaindosinensis* were the most frequently encountered species (28.74% of the sampled sites), followed by *Mongolodiaptomus botulifer* (24.14%), while *Neodiaptomus yangtsekiangensis*, *Tropodiaptomus oryzanus*, and *Tropodiaptomus* sp. are rare species found in a single locality. To date, 40 calanoid species (33 in the family Diaptomidae) have been recorded from Vietnam, and an updated list is presented. Seven species are potentially endemic to Vietnam. At the same sampling dates, the species richness of the calanoids was a range of 1–5 species per locality. The results of the Canonical Correspondence Analysis showed that pH and conductivity tended to be positively related to the calanoid distribution.

**Keywords:** Diaptomidae; distribution; endemic species; environmental variables; habitats; Mekong River; Southeast Asia; *Tropodiaptomus* sp.

## 1. Introduction

Copepods are microcrustaceans in the aquatic ecosystem and they play a vital role in energy transfer in marine and freshwater food chains. They are very abundant in freshwater, constituting a major component of most planktonic, benthic, and groundwater communities [1]. They serve mainly as primary consumers of phytoplankton [2], and as food for many larger invertebrates and vertebrates [3]. Copepods are very diverse and widely distributed. In freshwater, they can be found in a wide array of habitats such as lakes, ponds, swamps, rivers, canals, rice fields, and other subterrestrial environments [4]. In addition, copepods are good biological indicators, and can be used as tools to determine water quality [5], because the presence or absence of a species allows for deductions regarding the physical–chemical characteristics of the environment.

Over 13,000 copepod species are known globally, and the greatest diversity is found in the marine environment. Approximately 2800 species of free-living copepods are present in the freshwater ecosystem [1]. Calanoid copepods are the dominant animals of freshwater plankton worldwide. The Diaptomidae of the order Calanoida are the most species-rich and widespread calanoid family in inland waters, containing over 440 species in 62 genera. A total of 78 species of *Pseudodiaptomus* in the family Pseudodiaptomidae are most widely

distributed in marine and brackish habitats, and only four species are found in freshwater habitats [1,6,7]. In Southeast Asia (SEA), the diversity of freshwater diaptomid copepods has been well studied in Thailand, and 42 diaptomid species have been reported [8]. To date, other Southeast Asian countries have lower diversities of diaptomid species than Thailand [9–14].

Vietnam is located in the lower Mekong River Basin, together with Thailand, Laos, and Cambodia, between the latitudes of 8° and 24° north and the longitudes of 102° and 110° east. It covers an area of 331,210 km$^2$, and the land is mostly hilly, with dense forests [15]. Previous studies on freshwater copepods in Vietnam have mainly focused on taxonomy, diversity, and distribution [16,17] and most have been in the north, rather than in the southern areas [9,18–21]. Because of the differences in latitudes and the marked variety in topographical relief, the northern part consists mostly of highlands and the river delta, but the southern part of Vietnam is divided into coastal lowlands, mountains, and extensive forests [15]. The southern part of Vietnam has a tropical monsoon climate, with seasons differing only in the amount of precipitation and wind directions. There are two seasons: dry (November–April) and rainy (May–October) seasons. The objective of this study was to examine the diversity and distribution of calanoid copepods and their relationships to environmental variables in different water bodies in Southern Vietnam. In addition, the diversity of the diaptomid calanoids is reviewed and discussed.

## 2. Materials and Methods

### 2.1. Sample Collections

The study areas included freshwater sites in a municipality (Ho Chi Minh City) plus seven provinces (Bà Rịa–Vũng Tàu, Bình Dương, Bình Phước, Đồng Nai, Long An, Tây Ninh, and Tiền Giang) from the southeast and Mekong Delta of Southern Vietnam. Samples were collected from 87 sites (160 samples; 78 samples in the rainy season and 82 samples in the dry season) (Figure 1). Most of the sites were visited in both the rainy and dry seasons, but some could not be sampled during the rainy season because they were flooded. A variety of water habitats, including permanent water habitats (37 lakes, five ponds, 13 roadside canals, and 22 rivers) and temporary water habitats (10 rice fields) were sampled. Although most of the sampled habitats were freshwater, six sites were brackish waters.

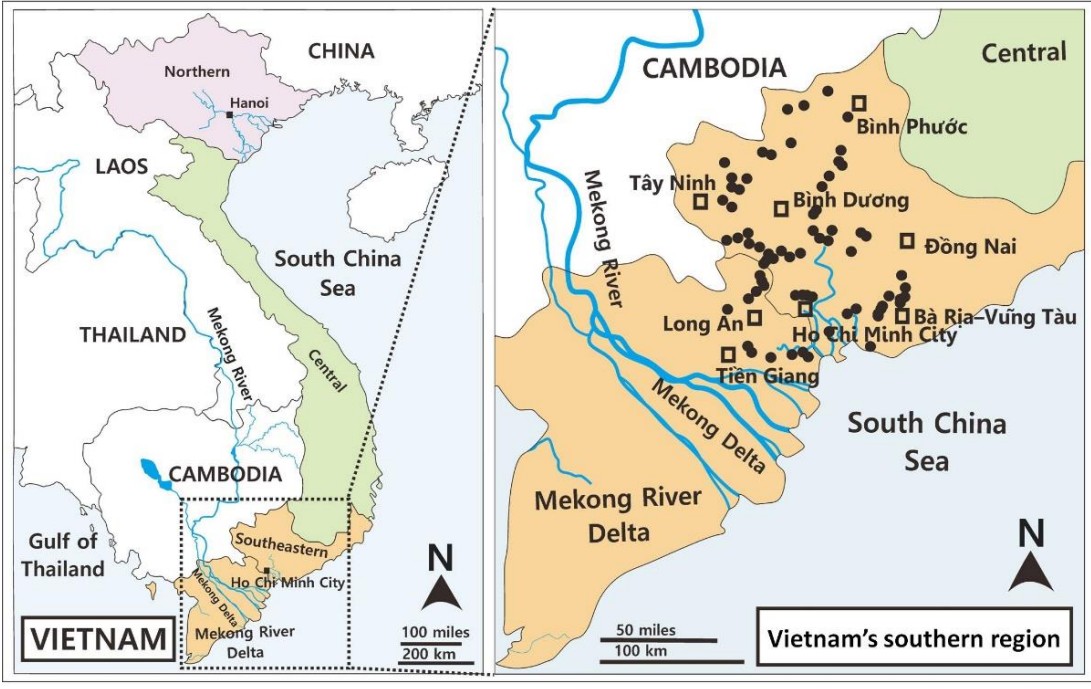

**Figure 1.** The map of southern Vietnam showing the locations of sampling sites (black dots).

Copepod samples were collected qualitatively during the dry (8–14 November 2012) and rainy (3–9 June 2013) seasons. Samplings were conducted using oblique tows from the shore with a 60 μm mesh size plankton net. Coordinates and altitude were measured using the Global Positioning System (GPS map Cx, Garmin). The habitat types, number of sites, coordinates, and altitudes of the sampling sites in each province are shown in Table 1. Some environmental variables, e.g., water temperature, pH, dissolved oxygen, and conductivity, were measured at each site using water quality multiparameter instruments (Horiba® U10), and the range values for all sites from the two seasons are shown in Table 2. Samples were preserved in 4% formalin and examined under a stereomicroscope (SZ-PT, Olympus). Photographs were taken using a compound microscope (SZX-ILLK 200, Olympus).

**Table 1.** Habitat types, number of sites, coordinate ranges, and altitudes of the sampling sites in eight provinces of southern Vietnam. The habitat types are represented as L = lake, P = pond, C = roadside canal, Ri = river, and Rf = rice field.

| Habitat Types | Number of Sites | Latitude (N) | Longitude (E) | Altitude (m) | Province |
|---|---|---|---|---|---|
| P, C, Ri, Rf | 15 | 10°39′–11°04′ | 106°20′–106°43′ | 5–18 | Ho Chi Minh City |
| L, P, Ri | 11 | 10°20′–10°43′ | 107°05′–107°18′ | 6–150 | Bà Rịa–Vũng Tàu |
| L, Ri | 7 | 10°58′–11°14′ | 106°38′–106°51′ | 9–47 | Bình Dương |
| L, Ri, Rf | 21 | 11°26′–11°58′ | 106°29′–106°58′ | 33–196 | Bình Phước |
| L | 5 | 10°46′–11°05′ | 106°59′–107°18′ | 36–189 | Đồng Nai |
| C, Ri, Rf | 7 | 10°45′–10°56′ | 106°22′–106°27′ | 7–11 | Long An |
| L, C, Ri, Rf | 13 | 11°06′–11°36′ | 106°10′–106°26′ | 11–43 | Tây Ninh |
| L, C, Ri, P | 8 | 10°21′–10°42′ | 106°19′–106°40′ | 4–23 | Tiền Giang |
| **Total** | **87** | | | | |

**Table 2.** Ranges of environmental variables measured from the sampled sites in southern Vietnam in the rainy and dry seasons.

| Environmental Variables | Rainy Season | Dry Season |
|---|---|---|
| Water temperature (°C) | 27.00–35.1 | 26.5–33.4 |
| pH | 5.1–8.7 | 5.1–8.0 |
| Dissolved oxygen (mg L$^{-1}$) | 1.3–10.0 | 1.9–10.6 |
| Conductivity (μS cm$^{-1}$) | 11–1980 | 25–3980 |

Specimens for scanning electron microscopy (SEM) were dehydrated in an ethanol series (50, 70, 80, 90, 95, 100, and 100%) for 15 min at each concentration. Specimens were dried in a critical-point dryer and were mounted on stubs using adhesive tape under a stereomicroscope. Dried specimens were coated with gold in a sputter-coater. The SEM photographs were taken using a scanning electron microscope (LEO, 1450VP).

*2.2. Data Analysis*

Standardized data were explored to detect outliers, then log10 [x + 1] transformed before performing statistical analyses to decrease the variance of the data set, and to avoid violating assumptions of normality. Data were analyzed using the PC-ORD version 5.0 program to explore correlations between calanoid species, habitats, and environmental variables, as well as between species and environmental variables. The similarity of the sites based on the presence of taxa was analyzed via cluster analysis, using the Jaccard similarity distance and Group Average Linkage method. Correlations between species and environmental variables were investigated in each habitat via a canonical correspondence analysis (CCA). The CCA procedure produces an ordination diagram in which species and habitats are represented by points and environmental variables by vectors. Monte Carlo permutation tests were performed to test the statistical strengths of eigenvalues of the ordination axes and species–environmental correlations. The frequency of species occurrence was calculated based on the cumulative number of all collected samples with

calanoid species. An independent samples t-test was utilized to compare the differences in environmental variables between the rainy and dry seasons. The differences in species numbers and the diversity between the two seasons were examined using non-parametric statistical methods. All statistical analyses were performed with SPSS 19.0 software. The significance level was 111, set as $p < 0.05$.

## 3. Results

### 3.1. Diversity of Calanoid Copepods in Southern Vietnam

A list of calanoid species identified from the samples examined, along with their habitat types, frequency percentages, and provinces where they occur in southern Vietnam, is provided in Table 3. Of the 13 taxa recorded, nine species are members of the family Diaptomidae, while three and one belonging to the families Pseudodiaptomidae and Acartiidae, respectively. The diversity of species among the eight genera is dominated by the genera *Mongolodiaptomus* and *Pseudodiaptomus*, with three known species each. Among these, four species (*Mongolodiaptomus malaindosinensis*, *M. mekongensis*, *Vietodiaptomus blachei*, and *Pseudodiaptomus siamensis*) (Figure 2A–E) are newly added to the fauna list of the country. An unidentified taxon, *Tropodiaptomus* sp. (Figure 2E), probably belongs to a new species. *Eodiaptomus draconisignivomi* and *M. malaindosinensis* were the most frequently encountered species, which were found at 28.7% of the sampled sites, followed by *Mongolodiaptomus botulifer* (24.1%), *Acartiella sinensis* (6.9%) (Figure 3), *Heliodiaptomus elegans* and *Pseudodiaptomus dauglishi* (5.7%), *Vietodiaptomus blachei*, *Pseudodiaptomus annandalei* and *Pseudodiaptomus siamensis* (3.4%), and *Mongolodiaptomus mekongensis* (2.3%). *Neodiaptomus yangtsekiangensis*, *Tropodiaptomus oryzanus*, and *Tropodiaptomus* sp. were infrequently found (1.1%), and only in a single locality.

**Table 3.** List of calanoid copepods recorded, with their habitat types, provinces where they occur, and frequency percentages. Frequency percentages were calculated from the cumulative number of the 87 sampling sites in Southern Vietnam. L = lake, P = pond, C = roadside canal, Ri = river, and Rf = rice field. Provinces are represented by numbers 1–8: 1 = Ho Chi Minh City, 2 = Bà Rịa–Vũng Tàu, 3 = Bình Dương, 4 = Bình Phước, 5 = Đồng Nai, 6 = Long An, 7 = Tây Ninh, and 8 = Tiền Giang.

| No | Species | Habitat Types | | | | | Provinces | Frequency (%) |
| | | Permanent Waters | | | | Temporary Waters | | |
| | | L | P | C | Ri | Rf | | |
|---|---|---|---|---|---|---|---|---|
| | Family Acartiidae Sars G.O., 1903 | | | | | | | |
| 1 | *Acartiella Sinensis* Shen and Lee, 1963 | | | √ | √ | | 1, 8 | 6.9 |
| | Family Diaptomidae Baird, 1850 | | | | | | | |
| 2 | *Eodiaptomus draconisignivomi* Brehm, 1952 | √ | √ | √ | √ | √ | 1, 2, 3, 4, 5, 6, 7 | 28.7 |
| 3 | *Heliodiaptomus elegans* Kiefer, 1935 | √ | | √ | √ | √ | 1, 4, 7 | 5.7 |
| 4 | *Mongolodiaptomus botulifer* (Kiefer, 1974) | √ | | √ | √ | √ | 1, 2, 4, 5, 6, 7, 8 | 24.1 |
| 5 | *Mongolodiaptomus malaindosinensis* (Lai and Fernando, 1978) * | √ | √ | √ | √ | √ | 1, 3, 4, 5, 6, 7, 8 | 28.7 |
| 6 | *Mongolodiaptomus mekongensis* Sanoamuang and Watiroyram, 2018 * | √ | | | | √ | 4 | 2.3 |

Table 3. *Cont.*

| No | Species | Permanent Waters | | | | Temporary Waters | Provinces | Frequency (%) |
|---|---|---|---|---|---|---|---|---|
| | | **Habitat Types** | | | | | | |
| | | L | P | C | Ri | Rf | | |
| 7 | *Neodiaptomus yangtsekiangensis* Mashiko, 1951 | | | | | √ | 7 | 1.1 |
| 8 | *Tropodiaptomus oryzanus* Kiefer, 1937 | | | | | √ | 4 | 1.1 |
| 9 | *Tropodiaptomus* sp. | | | | | √ | 1 | 1.1 |
| 10 | *Vietodiaptomus blachei* (Brehm, 1951) * | √ | | | | √ | 4 | 3.4 |
| | Family Pseudodiaptomidae Sars G.O., 1902 | | | | | | | |
| 11 | *Pseudodiaptomus annandalei* Sewell, 1919 | √ | √ | | √ | | 2 | 3.4 |
| 12 | *Pseudodiaptomus dauglishi* Sewell, 1932 | √ | | √ | √ | | 3, 5, 8 | 5.7 |
| 13 | *Pseudodiaptomus siamensis* Srinui, Nishida and Ohtsuka, 2013 * | | | √ | √ | | 1 | 3.4 |
| | **Total** | 8 | 3 | 7 | 8 | 9 | | |

* = new record for Vietnam.

Moreover, *E. draconisignivomi* and *M. malaindosinensis* were widely distributed and found in all habitat types, while *N. yangtsekiangensis*, *T. oryzanus*, and *Tropodiaptomus* sp. were restricted to rice fields. Likewise, *E. draconisignivomi*, *M. botulifer*, and *M. malaindosinensis* were widely distributed across 7–8 provinces of Southern Vietnam (Table 3). The species richness showed little difference between the dry season (10 species) and the rainy season (9 species). *E. draconisignivomi*, *H. elegans*, *M. botulifer*, *M. malaindosinensis*, *P. annandalei*, and *P. dauglishi* were found in the two seasons. *M. mekongensis*, *N. yangtsekiangensis*, *T. oryzanus*, and *V. blachei* were only found during the dry season, whereas *A. sinensis*, *P. siamensis*, and *Tropodiaptomus* sp. were only found during the rainy season (Figure 4). Species that were found with high frequency in both the dry and rainy seasons were *E. draconisignivomi* (20.7% and 16.1%, respectively), followed by *M. malaindosinensis* (19.5, 18.4%), and *M. botulifer* (17.2, 11.5%), respectively.

At the same sampling dates, the species richness of the diaptomid calanoids ranged from 1 to 5 species per site. Most of the sampled sites contained 1 to 2 species, but only one site (a rice field in Ho Chi Minh City) had five species co-occurring during the rainy season (*E. draconisignivomi*, *H. elegans*, *M. botulifer*, *M. malaindosinensis*, and *Tropodiaptomus* sp.).

### 3.2. The Relationship between Calanoid Species and Environmental Variables

The similarity of the sites is based on the presence of the calanoid copepods. Hierarchical analysis recognized five habitats at a Jaccard similarity distance of 100%. Based on a cluster analysis, there are three groups of habitats: (1) river and roadside canal (IIIA) and lake (IIIB); (2) rice field (IIB); and (3) pond (IB) (Figure 5). The Jaccard's index revealed a closer relationship between the habitats of river and roadside canal, resulting in a similarity percentage of 98.75%. Seven species (*Eodiaptomus draconisignivomi*, *Mongolodiaptomus malaindosinensis*, *Heliodiaptomus elegans*, *M. botulifer*, *Pseudodiaptomus dauglishi*, *P. siamensis*, and *Acartiella sinensis*) were found in this group. Group IIIB, consisting only of lakes, showed approximately 81% similarity in species composition to Group IIIA. Group IIB consists only of rice fields, and showed an approximately 46% similarity in species composition to Group IIA, and all

diaptomid calanoids (nine species) were found in this group. In contrast, in Group IB, pond habitat was separated from lower species richness. Three species (*E. draconisignivomi*, *M. malaindosinensis,* and *Pseudodiaptomus annandalei*) were found in this group. Based on the diagram of species composition, most of the calanoids were found in both permanent and temporary water habitats. However, *Neodiaptomus yangtsekiangensis*, *Tropodiaptomus oryzanus*, and *Tropodiaptomus* sp. were found only in temporary water habitats, while the three taxa of the genus *Pseudodiaptomus* and *A. sinensis* were found only in permanent water habitats.

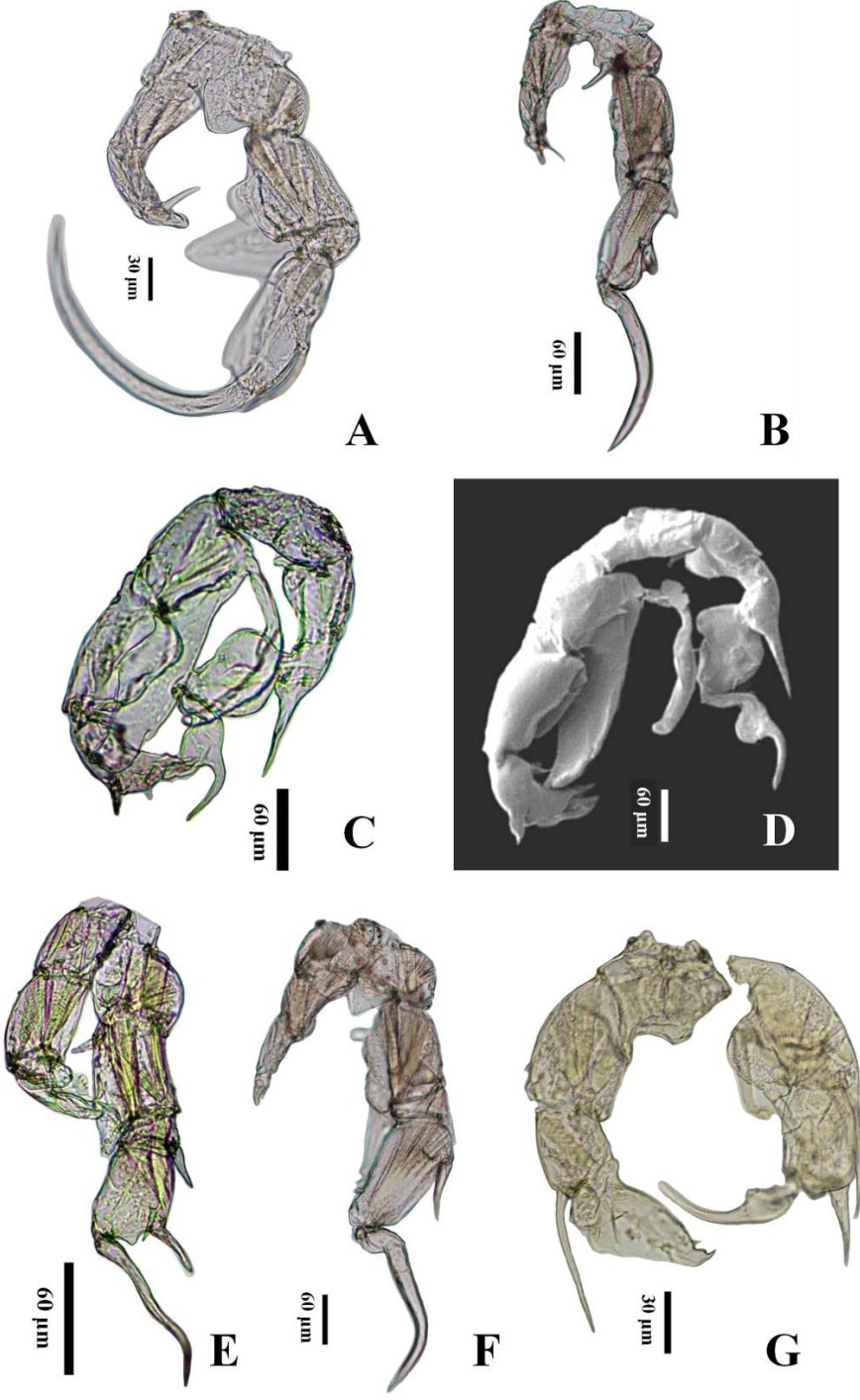

**Figure 2.** Photographs of the male leg 5 (P5) of calanoid copepods recorded in Vietnam: *Mongolodiaptomus malaindosinensis* (**A**), *Mongolodiaptomus mekongensis* (**B**), *Pseudodiaptomus siamensis* (**C**,**D**), *Tropodiaptomus* sp. (**E**), *Vietodiaptomus blachei* (**F**), and *Pseudodiaptomus dauglishi* (**G**).

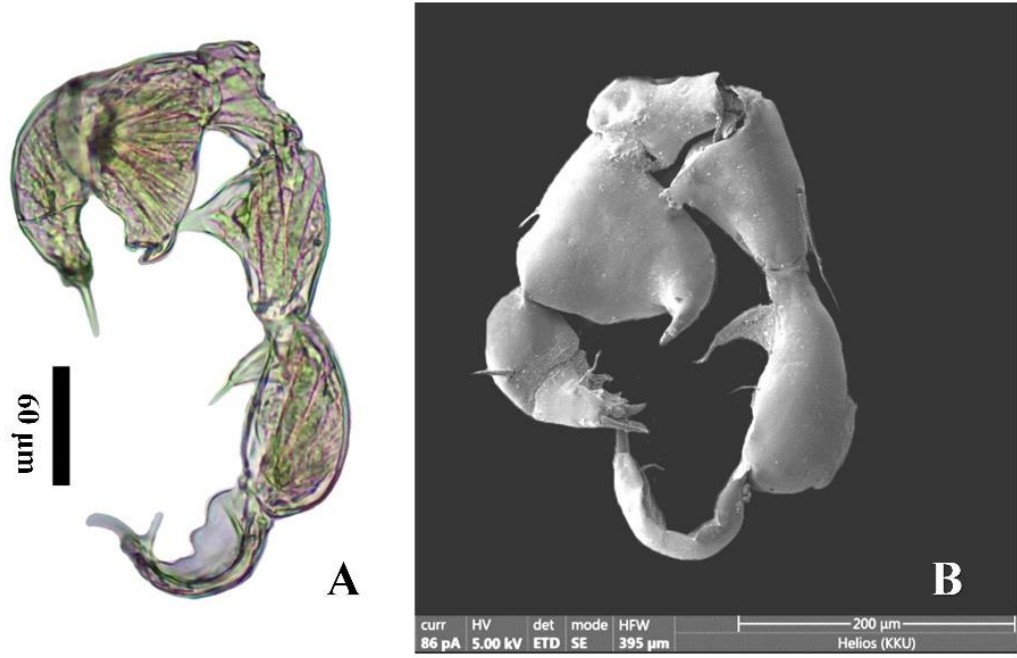

**Figure 3.** Photographs of the male leg 5 (P5) of a brackish-water calanoid copepod, *Acartiella sinensis*: pictures taken using a light microscope (**A**) and a scanning electron microscope (**B**).

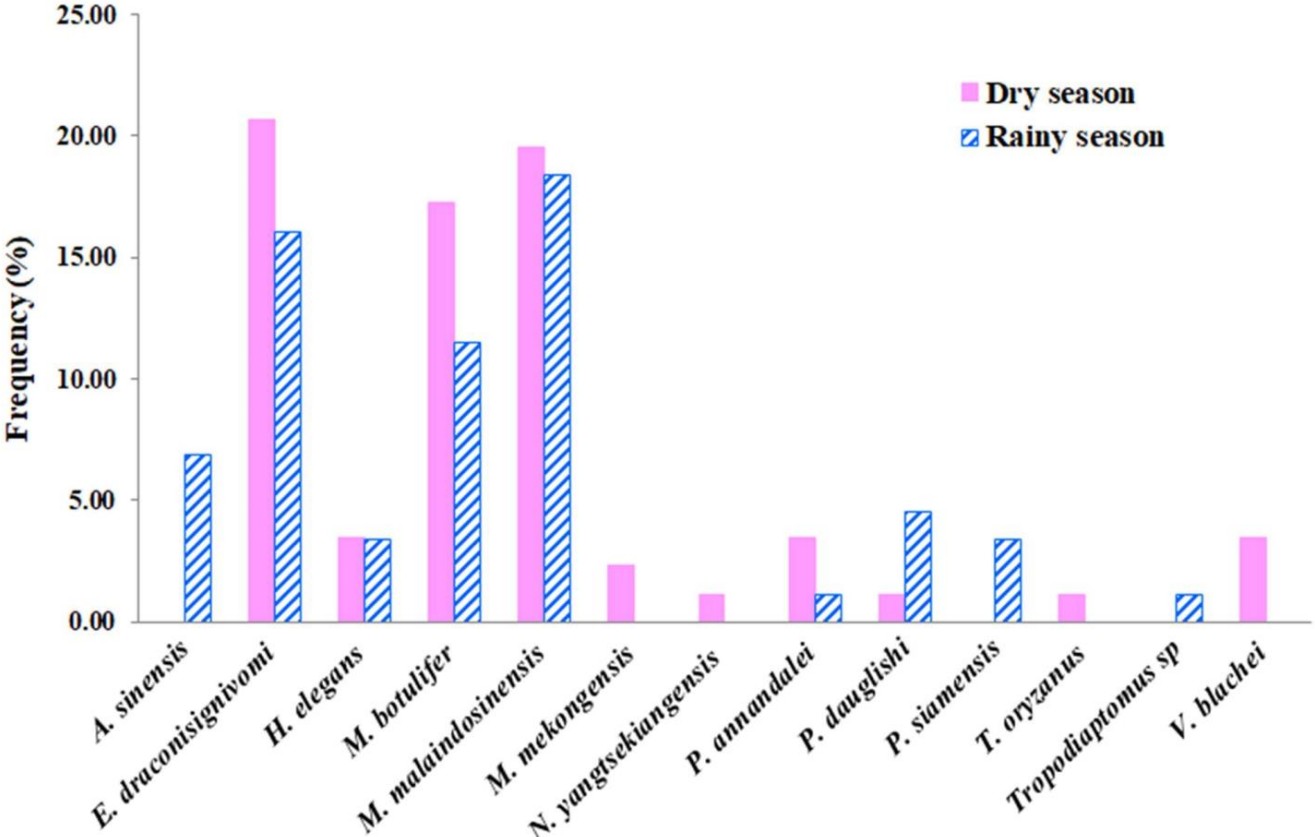

**Figure 4.** The frequency percentages of calanoid species found in southern Vietnam during the dry and rainy seasons.

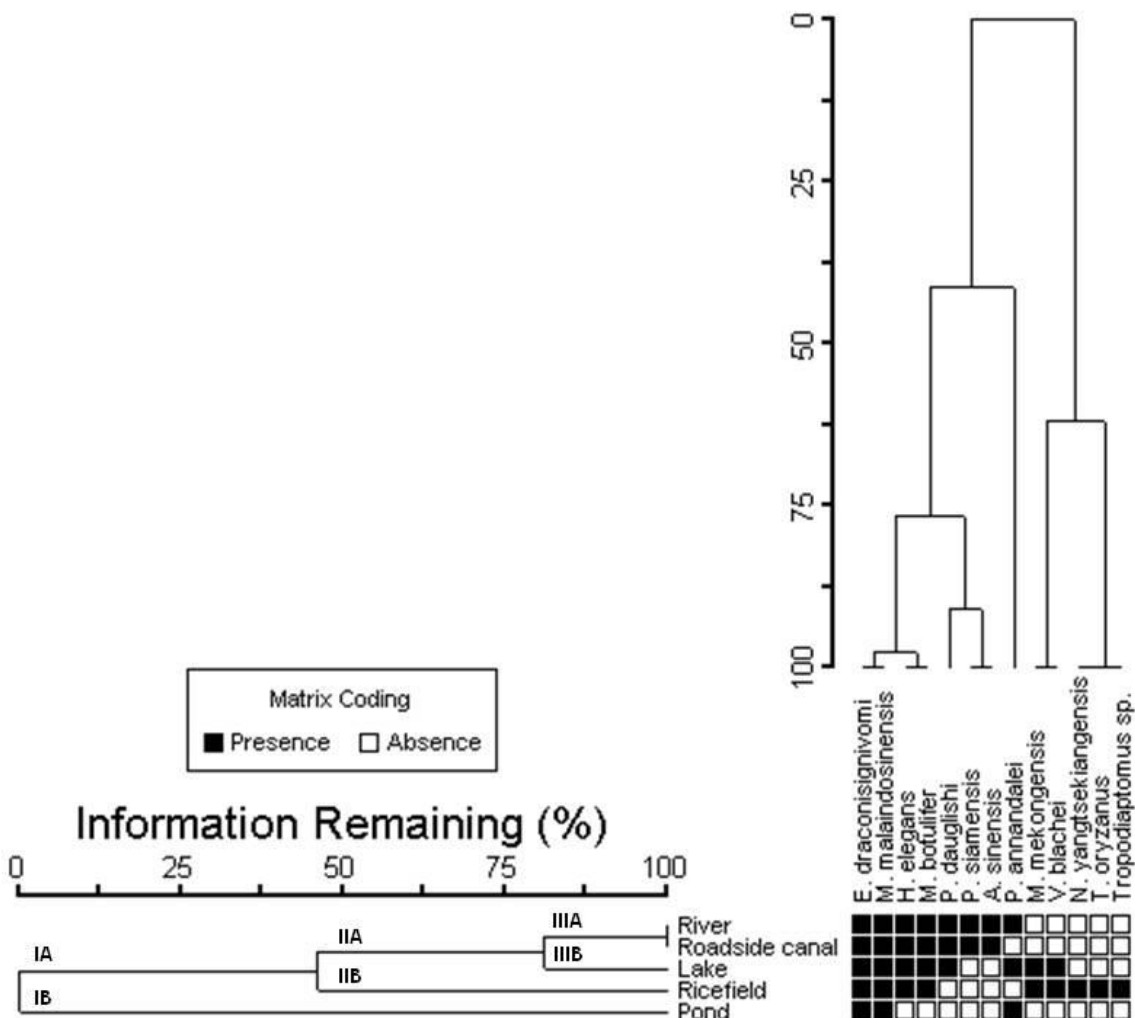

**Figure 5.** The similarity of the sites is based on the species present in each of the different habitats in southern Vietnam.

Some environmental parameters associated with water quality in different habitats of southern Vietnam were measured in both the rainy and dry seasons (Table 2). The independent samples t-test indicated no significant differences between the rainy and dry seasons regarding water environmental variables ($p > 0.05$). The temperature of the water varied from 26.5 to 35.1 °C, the pH varied from 5.1 to 8.7, the dissolved oxygen varied from 1.3 to 10.6 mg L$^{-1}$, and the conductivity varied from 11 to 3980 μS cm$^{-1}$.

The first two axes of CCA ordination accounted for 85% of the variance in the species data (Table 4). The eigenvalues for axis 1 and axis 2 were 0.430 and 0.207, respectively. The correlation (r-value) between the species and environmental variables for axis 1 and 2 was the same value of 1.000. The percentage of variance explained by axes 1 and 2 was 57.4% and 27.6%, respectively. The Monte Carlo test showed that the environmental variables were significantly correlated ($p = 0.036$) (Figure 6). Axes 1 and 2 were positively correlated to water temperature, conductivity; and pH and dissolved oxygen, respectively. The CCA divided the collected species into four groups; i.e., Group 1, copepods that are likely to be found in habitats with higher-than-average water temperatures (r = 0.766), dissolved oxygen (r = 0.782), and pH (r = 0.844), were *M. mekongensis* and *Vietodiaptomus blachei*. Group 2, copepods that are likely to be found in habitats with lower water temperatures, dissolved oxygen, and pH, were *A. sinensis*, *P. siamensis*, *P. dauglishi*, *H. elegans*, and *M. botulifer*. In Group 3, copepods that are likely to be found with higher-than-average conductivities (r = 0.655) were *P. annandalei*, *E. draconisignivomi*, and *M. malaindosinensis*. In contrast,

*N. yangtsekiangensis*, *T. oryzanus*, and *Tropodiaptomus* sp. were found in Group 4 copepods with low conductivity.

**Table 4.** Canonical correspondence analysis for environmental variables. (**a**) Axis summary statistics for the three extracted canonical axes, as well as the percentage of variance explained by CCA ordination. (**b**) The relationship between environmental variables and ordination axes.

|  | Axis 1 | Axis 2 | Axis 3 |
|---|---|---|---|
| Total variance in the species data: 0.7496 | | | |
| (a) Axis summary statistics and variance in species data | | | |
| Eigenvalue | 0.430 | 0.207 | 0.102 |
| Variance in species data | | | |
| % of variance explained | 57.4 | 27.6 | 13.7 |
| cumulative % explained | 57.4 | 84.9 | 98.6 |
| Pearson correlation, Spp–Envt | 1.000 | 1.000 | 1.000 |
| (b) Correlations of environmental parameters and canonical axes | | | |
| Temperature | −0.766 | 0.190 | −0.060 |
| pH | −0.047 | 0.844 | 0.523 |
| Conductivity | 0.655 | 0.352 | −0.668 |
| DO | −0.391 | 0.782 | 0.485 |

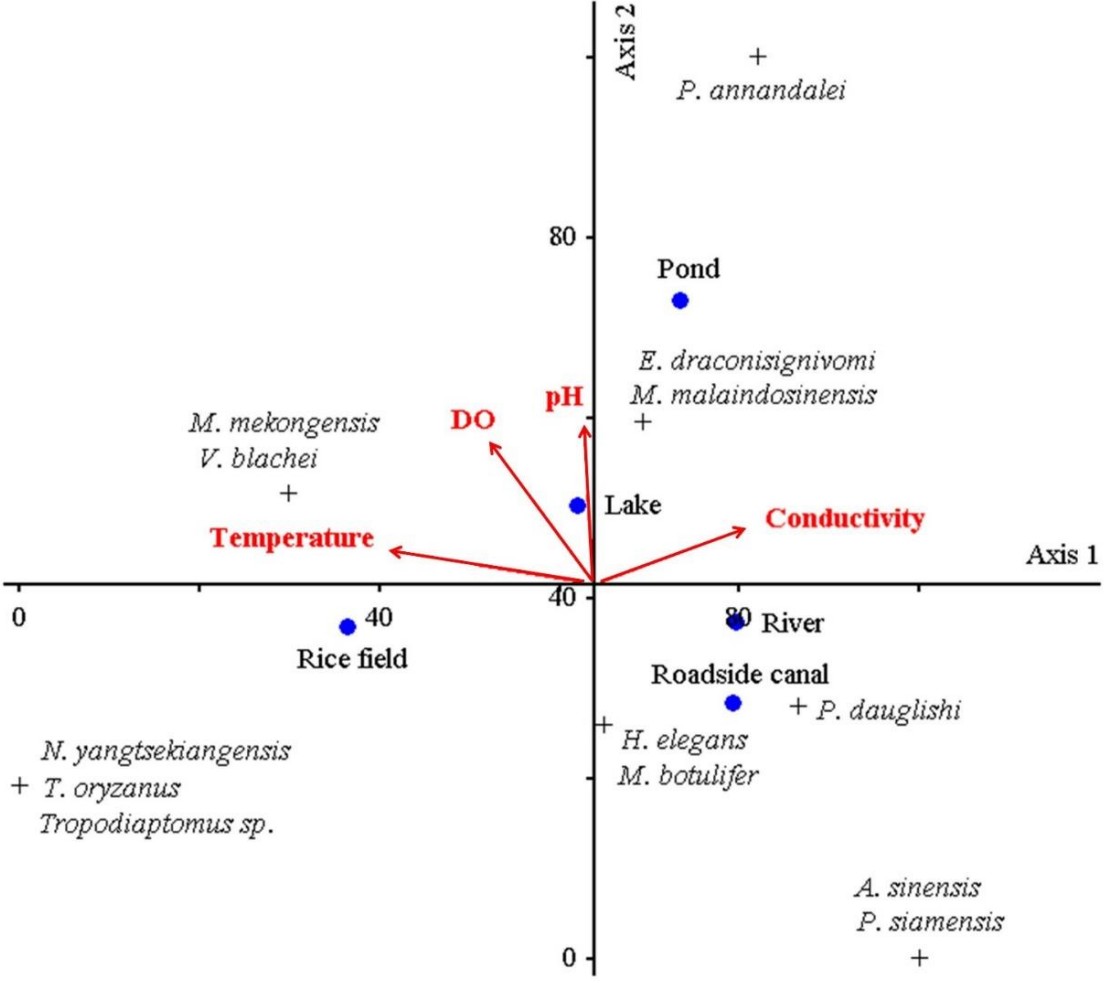

**Figure 6.** Ordination diagram based on canonical correspondence analysis (CCA) of copepod species scores concerning environmental variables (the Monte Carlo test; *p* = 0.036, Eigenvalue axis 1 = 0.419, Eigenvalue axis 2 = 0.187). Filled circles represent habitats, plus symbols represent species, and arrowed lines represent environmental variables.

Based on the cluster and the CCA analyses, the calanoid species can be classified into four groups:

Group I: Taxa with wide distribution in the southern part of Vietnam, which can be found in both permanent and temporary waters; i.e., *E. draconisignivomi*, *M. malaindosinensis*, *H. elegans*, and *M. botulifer*.

Group II: Taxa with restricted distribution in the southern part of Vietnam, which can be found in habitats with high water temperatures, dissolved oxygen, and pH; i.e., *M. mekongensis* and *V. blachei*.

Group III: Taxa with restricted distribution in the southern part of Vietnam, which can be found in temporary water habitats with low conductivity; i.e., *N. yangtsekiangensis*, *T. oryzanus*, and *Tropodiaptomus* sp.

Group IV: Taxa with wide distribution in the southern part of Vietnam, which can be found in brackish and seawater with a high conductivity of over 3000 µS cm$^{-1}$; i.e., *A. sinensis*, *P. annandalei*, *P. siamensis*, and *P. dauglishi* This group was found in permanent water habitats, especially in rivers that are connected to the sea.

## 4. Discussion

### 4.1. Calanoid Copepod Diversity in Vietnam

The results of our study update on the species richness of freshwater calanoid copepods in Vietnam recorded from 35 [9,21] to 40 species (Table 5). The species diversity of the calanoids from this country is dominated by members of the family Diaptomidae, similar to that from Thailand and other Southeast Asian countries [8,22]. The number of diaptomid species recorded (33 species) is lower than that found in Thailand (42 species) [8], but higher than the other Southeast Asian countries, which range from 8 to 24 species (Table 6). The diversity of species among the 13 genera of Diaptomidae is dominated by the genus *Mongolodiaptomus*, with 10 known species, followed by *Tropodiaptomus* and *Neodiaptomus*, with four species each. The numbers of species in the dominant genera in Vietnam correspond well with those reported in Thailand [8], except that *Phyllodiaptomus* consists of one species in Vietnam, but six species in Thailand [8,23]. However, two subterranean diaptomid species (*Hadodiaptomus dumonti* and *Nannodiaptomus phongnhaensis*) have been recorded in Vietnam [24,25], while there is no record of subterranean species in Thailand yet.

**Table 5.** Updated list of calanoid copepods recorded in Vietnam.

| | Species | This Study | Phan et al. (2015) [21] | Tran et al. (2016) [9] |
|---|---|---|---|---|
| | Family Acartiidae | | | |
| 1 | *Acartiella sinensis* Shen and Lee, 1963 | √ | √ | |
| | Family Diaptomidae | | | |
| 2 | *Allodiaptomus mieni* Dang and Ho, 1985 | | √ | √ |
| 3 | *A. raoi* Kiefer, 1936 | | √ | √ |
| 4 | *A. specillodactylus* Shen and Tai, 1964 | | | √ |
| 5 | *Dentodiaptomus javanus* (Grochmalicki, 1915) | | | √ |
| 6 | *Dolodiaptomus spinicaudatus* Shen and Tai, 1964 | | | √ |
| 7 | *Eodiaptomus draconisignivomi* Brehm, 1952 | √ | √ | √ |
| 8 | *Hadodiaptomus dumonti* Brancelj, 2005 | | | √ |
| 9 | *Heliodiaptomus elegans* Kiefer, 1935 | √ | √ | √ |
| 10 | *H. falxus* Shen and Tai, 1964 | | | √ |
| 11 | *Mongolodiaptomus birulai* (Rylov, 1923) | | | √ |
| 12 | *M. botulifer* (Kiefer, 1974) | √ | √ | √ |
| 13 | *M. calcarus* Shen and Tai, 1965 | | | √ |
| 14 | *M. gladiolus* (Shen and Lee, 1963) | | √ | √ |
| 15 | *M. malaindosinensis* (Lai and Fernando, 1978) | √ | | |
| 16 | *M. mekongensis* Sanoamuang and Watiroyram, 2018 | √ | | |
| 17 | *M. pectinidactylus* (Shen and Tai, 1964) | | | √ |
| 18 | *M. uenoi* (Kikuchi, 1936) | | | √ |
| 19 | *Mongolodiaptomus* sp. 1 | | | √ |

**Table 5.** *Cont.*

| | Species | This Study | Phan et al. (2015) [21] | Tran et al. (2016) [9] |
|---|---|---|---|---|
| 20 | *Mongolodiaptomus* sp. 2 | | | √ |
| 21 | *Nannodiaptomus phongnhaensis* Dang and Ho, 2001 | | | √ |
| 22 | *Neodiaptomus curvispinosus* Dang and Ho, 2001 | | | √ |
| 23 | *N. schmackeri* (Poppe and Richard, 1892) | | | √ |
| 24 | *N. vietnamensis* Dang and Ho, 1998 | | | √ |
| 25 | *N. yangtsekiangensis* Mashiko, 1951 | √ | √ | √ |
| 26 | *Phyllodiaptomus tunguidus* Shen and Tai, 1964 | | | √ |
| 27 | *Sinodiaptomus sarsi* (Rylov, 1923) | | | √ |
| 28 | *Tropodiaptomus foresti* Defaye, 2002 | | | √ |
| 29 | *T. oryzanus* Kiefer, 1937 | √ | √ | √ |
| 30 | *T. vicinus* (Kiefer, 1930) | | √ | √ |
| 31 | *Tropodiaptomus* sp. | √ | | |
| 32 | *Vietodiaptomus blachei* (Brehm, 1951) | √ | | |
| 33 | *V. hatinhensis* Dang, 1977 | | √ | √ |
| 34 | *V. tridentatus* Dang and Ho, 1985 | | √ | √ |
| | Family Pseudodiaptomidae | | | |
| 35 | *Pseudodiaptomus annandelei* Sewell, 1919 | √ | √ | |
| 36 | *P. dauglishi* Sewell, 1932 | √ | √ | |
| 37 | *P. incisus* Shen and Lee, 1963 | | √ | |
| 38 | *P. inopinus* Burckhardt, 1913 | | √ | |
| 39 | *P. siamensis* Srinui, Nishida and Ohtsuka, 2013 | √ | | |
| 40 | *P. trihamatus* Wright, 1937 | | √ | |

**Table 6.** The number of diaptomid species found in some Southeast Asian countries.

| Country | Number of Species | Reference |
|---|---|---|
| Thailand | 42 | Sanoamuang and Dabseepai, 2021 [8] |
| Vietnam | 33 | Tran et al., 2016 [9]; this study |
| Cambodia | 24 | Chaichareon, 2011 [10] |
| Laos | 19 | Sanoamuang and Watiroyram, 2019 [11] |
| Indonesia | 17 | Alekseev et al., 2013 [13] |
| Malaysia and Singapore | 12 | Lim and Lai, 2014 [12] |
| Philippines | 8 | Lopez et al., 2017 [14] |

Of the 33 diaptomid species recorded in Vietnam, 14 (42.4%) belong to species occurring in the lower Mekong River Basin (Thailand, Cambodia, Laos, and Vietnam) (Table 7). Seventeen species (51.5%) are the same as those recorded in China [26]. Fifteen species (45.5%) are the same as those found in Thailand (see Tables 2 and 4 in Sanoamuang and Dabseepai, 2021 [8]). Seven species are potentially endemic to Vietnam (Table 7), namely: *Allodiaptomus mieni*, *Hadodiaptomus dumonti*, *Nannodiaptomus phongnhaensis*, *Vietodiaptomus hatinhensis*, *Neodiaptomus curvispinosus*, *Neodiaptomus vietnamensis*, and *Tropodiaptomus foresti*. *Allodiaptomus specillodactylus*, *Dolodiaptomus spinicaudatus*, *Heliodiaptomus falxus*, and *Mongolodiaptomus gladiolus* have, to date, been found only in Vietnam and South China [9,26], while *Vietodiaptomus tridentatus* has so far been found only in Vietnam and Laos ([9], Sanoamuang, unpublished data).

The possible new species, *Tropodiaptomus* sp., is closely similar to *Tropodiaptomus ruttneri* (Brehm, 1923), particularly in the structure of the second exopodal segment (Exp2) of the male right leg 5. The Exp2 of both species has a trapezoid-like shape, with a principal lateral spine located approximately two-thirds of the length of the segment. However, the male right leg 5 of *Tropodiaptomus* sp. is different from *T. ruttneri* in various characteristics: (1) the first exopodal segment (Exp1) in *Tropodiaptomus* sp. has a long, prominent spiniform

process on the distal outer margin, whereas *T. ruttneri* has a small spiniform process; (2) the Exp2 in *Tropodiaptomus* sp. has one slender-shaped, accessory lateral spine situated close to the principal lateral spine, whereas *T. ruttneri* has two tiny knobs on the disto-outer margin; and (3) the basis in *Tropodiaptomus* sp. has a spiniform process inserted on the proximal third of the inner margin, whereas *T. ruttneri* has two small knobs. *Tropodiaptomus* sp. is rare, and only two male specimens were collected. Thus, a morphological description of this species will be published when we will have more specimens at disposition.

**Table 7.** Diaptomid copepods from Vietnam are classified according to their geographic distribution. The Lower Mekong River Basin region consists of Thailand, Myanmar, Laos, Cambodia, and Vietnam. The Southeast Asian region consists of Thailand, Myanmar, Laos, Cambodia, Vietnam, Malaysia, Singapore, the Philippines, Indonesia, and Brunei. The East Asian region consists of China, Japan, North Korea, South Korea, Mongolia, and Taiwan. The South Asian region consists of India, Sri Lanka, Afghanistan, Bangladesh, Bhutan, Nepal, and Pakistan.

| Distributional Regions | Number of Species | List of Species |
|---|---|---|
| Vietnam only; potentially endemic species | 7 | *Allodiaptomus mieni*, *Hadodiaptomus dumonti*, *Nannodiaptomus phongnhaensis*, *Vietodiaptomus hatinhensis*, *Neodiaptomus curvispinosus*, *N. vietnamensis*, *Tropodiaptomus foresti* |
| Vietnam and South China only | 4 | *Allodiaptomus specillodactylus*, *Dolodiaptomus spinicaudatus*, *Heliodiaptomus falxus*, *Mongolodiaptomus gladiolus* |
| Vietnam and Laos only | 1 | *Vietodiaptomus tridentatus* |
| Lower Mekong River Basin | 2 | *Eodiaptomus draconisignivomi*, *Mongolodiaptomus mekongensis* |
| Southeast Asia | 3 | *Mongolodiaptomus botulifer*, *M. malaindosinensis*, *Vietodiaptomus blachei* |
| Southeast Asia and East Asia | 10 | *Allodiaptomus raoi*, *Dentodiaptomus javanus*, *Mongolodiaptomus birulai*, *M. calcarus*, *M. pectinidactylus*, *M. uenoi*, *Neodiaptomus yangtsekiangensis*, *Phyllodiaptomus tunguidus*, *Sinodiaptomus sarsi*, *Tropodiaptomus oryzanus* |
| Southeast Asia, East and South Asia | 3 | *Heliodiaptomus elegans*, *Neodiaptomus schmackeri*, *Tropodiaptomus vicinus* |
| Total | 30 | |

Note: Three unidentified species in Table 5 (*Mongolodiaptomus* sp. 1, *Mongolodiaptomus* sp. 2, and *Tropodiaptomus* sp.) are not included in this table.

A criterion relating to the armature details of the Exp2 of the male right P5 has been proposed to differentiate the genus *Mongolodiaptomus* from the close congeners *Neodiaptomus* and *Allodiaptomus* [27]. However, the taxonomic status of some species recorded from Vietnam by Tran et al. (2016) [9], e.g., *A. mieni*, *M. gladiolus*, and the two unidentified *Mongolodiaptomus*, may need to be re-identified using both morphological and molecular characters.

Based on this study, *Eodiaptomus draconisignivomi*, *Mongolodiaptomus malaindosinensis*, and *M. botulifer* are the species that occur in a variety of habitats and are the most frequently encountered species in the sampled sites. Similarly, these species are recorded as being common and the most frequently encountered in Thailand [8]. However, for the genus *Mongolodiaptomus*, *M. malaindosinensis* is the most frequently encountered species in southern Vietnam, while *M. botulifer* is the most common species in Thailand. The records for *M. mekongensis* and *T. oryzanus* are rare in occurrence in this study, but are uncommon

in Thailand [8]. To date, *M. mekongensis* has only been found in the four countries of the lower Mekong River Basin: Thailand, Laos, Cambodia, and Vietnam [28]. In contrast, *N. yangtsekiangensis* and *V. blachei* are rare in southern Vietnam, but they are common in Thailand [8].

The species richness of the diaptomid calanoids at the same sampling dates in Southern Vietnam was a range of 1–5 species per locality, and 1–2 species usually co-occurred in the same location, while that in Thailand, it was within a range of 1–7, and 2–3 species usually coexisted in the same location [8].

In the current study, we have also added information regarding the species richness and distribution of non-diaptomid calanoids in Vietnam. Seven species of *Pseudodiaptomus* and one species of *Acartiella* have so far been recorded. Members of the family Acartiidae have previously been reported only in brackish and seawater [7]. However, in this study, *Acartiella sinensis* was found in the rainy season in freshwater habitats (rivers and roadside canals in Tiền Giang province and Ho Chi Minh City) that are connected to the sea. According to previous data, this species occurs in brackish water and is quite common in the Mekong Delta region [21]. Likewise, members of the most common genus, *Pseudodiaptomus*, in the family Pseudodiaptomidae, are widely distributed in brackish and seawater, although a few species have been found in freshwater habitats [7]. For the first time in Vietnam, we reported *Pseudodiaptomus siamensis*. It was found in roadside canals and tributaries that connected to the Mekong River in Ho Chi Minh City. This species was recently described from specimens collected in the Prasae river estuary, on the eastern coast of the Gulf of Thailand [29]. In addition, *Pseudodiaptomus annandelei* was found in a wide range of habitats, including freshwater habitats such as rivers, lakes, and ponds, which correspond with the previous records of this species in fresh and brackish water in the Mekong Delta region by Phan et al. (2015) [21].

A comparison of the diaptomid species richness among regions reveals that of the 33 species recorded from the whole country, the Northern region is the most diverse, with 16 species. While the Central coast and Mekong River delta have 12 species each, the Central Highlands has 11 species [9]. From this study, we have updated the number of species found in the South from 7 to 11. The lower species richness (nine species) from our study may be due to the fact that our sampling area only covers approximately one-fifth of the country. We need to collect more samples from the vast area of the Mekong River Delta and other regions to obtain more species. Unfortunately, due to rapid economic development in Vietnam, the natural water quality in the southern region has been contaminated with organic substances, nutrients, and some heavy metals such as Cd, Hg, and As. The quality of the water, particularly in Dong Nai, Long An and Ho Chi Minh City, is the most polluted [30].

*4.2. The Relationship between Calanoid Species and Environmental Variables*

Our preliminary results on the similarity of the 87 sites in southern Vietnam show three groups of habitats (1. river–roadside canal–lake, 2. rice field, and 3. pond). Based on the diagram of species composition, most of the calanoids were found in both permanent and temporary water habitats. *N. yangtsekiangensis* and *T. oryzanus* were found only in temporary water habitats in southern Vietnam, but they were reported in both permanent and temporary water habitats in Thailand [8], and only in permanent water habitats in Cambodia [10].

It is known that the species richness and composition of the zooplankton assemblages are affected by the physical factors of the water bodies. Changes in community structure in freshwater habitats are determined by the physical environmental factors that limit the potential breadth of species distribution and the biotic interactions that determine the success of the species [31]. In this study, the species associated with high conductivity are *A. sinensis*, *P. annandelei*, *P. siamensis*, and *P. dauglishi*. Our results showed that *Pseudodiaptomus* species were positively related with conductivity because they are able to live in marine and brackish as well as freshwater, which corresponds to previous records documented by Walter and Boxshall (2021) [7]. In contrast, *N. yangtsekiangensis*, *T. oryzanus*,

and *Tropodiaptomus* sp. are negatively related with conductivity, so that they are found in freshwater habitats only. The records for *M. mekongensis* and *V. blachei* were rare in southern Vietnam. Both species were found in habitats with high dissolved oxygen (7.0−7.8 mg L$^{-1}$) and pH (7.3−7.5), only in Binh Phuoc province. Previous research in Southern Vietnam found that the water quality index in Binh Phuoc is good, with dissolved oxygen and pH of around 7.0 and 7.2, respectively [30]. In comparison to other provinces with crowded industrial parks, especially Dong Nai, Ho Chi Minh City, and Long An, low DO and pH were recorded but *M. mekongensis* and *V. blachei* were not found. It seems that both species can be indicators of good freshwater sources and with less nutrient contamination. The species richness of calanoids in Southern Vietnam showed little difference between the dry and rainy seasons, which is in accordance with that of Thailand [32], Cambodia [10], and Laos [33]. This can be explained by the fact that the water temperature and other variables are not significantly different among the seasons (Table 2).

## 5. Conclusions

Vietnam can be considered to be a biodiversity hotspot for diaptomid copepods in Southeast Asia. Its species richness may increase with intensive surveys, particularly in the remote subterranean areas. The species composition of the diaptomid copepods of Vietnam is similar to those found in neighboring countries in the lower Mekong River Basin and southern China. Seven species are potentially endemic to Vietnam. Studies on the effects of eutrophication and water pollution on the species diversity of freshwater copepods may help us to promote an awareness of biological conservation. We hope that this research will contribute to our understanding of calanoid diversity in Vietnam and lead to more phylogenetic, ecological, evolutionary, and molecular biology studies.

**Author Contributions:** P.B. contributed to the literature review, sample collection and identification, analyzing data, figure preparation, and writing the early drafts of the manuscript. L.S. contributed to the study's concept and design, literature review, and writing and editing the final manuscript. All authors have read and agreed to the published version of the manuscript.

**Funding:** This research was funded by a grant from Khon Kaen University's Post-Doctoral Training Program (PD2563-11).

**Institutional Review Board Statement:** Not applicable.

**Informed Consent Statement:** Not applicable.

**Data Availability Statement:** The data presented in this study are available in the article.

**Acknowledgments:** The authors would like to thank Mau Trinh-Dang (University of Da Nang, Vietnam) and Weerathan Maskasem for their assistance in the sample collections. For their technical help in the laboratory, Supattra Tiang-nga, Kamonwan Komput, Nattaporn Plangklang, and Prapat-sorn Dabseepai are acknowledged. Additionally, Nithikarn Sanoamuang is thanked for linguistic corrections to the manuscript. Helpful comments and suggestions made by the reviewers are very much appreciated. This work has no ethical approval code because the research was conducted in 2012–2013, prior to the announcement of the Thai Law on Animals for Scientific Purposes, 2015.

**Conflicts of Interest:** The authors declare no conflict of interest.

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
