# Peer review of "Diversity of Freshwater Calanoid Copepods (Crustacea: Copepoda: Calanoida) in Southern Vietnam with an Updated Checklist for the Country"

_diversity, doi:10.3390/d14070523_

Round 1
Reviewer 1 Report
I believe that this valuable manuscript is acceptable for publication because the findings of this study offer new data concerning the studied subject and obviously the results are novel for the regional and international literature.
Author Response
Please see the attached file, Thank you.

Reviewer 2 Report
The study of regional species diversity is conducive to the understanding of greater regional biodiversity. This paper studied the diversity of freshwater calanoid copepods from different habitats in southern Vietnam, and analyzed the whole calanoid copeods in Vietnam. The calanoid distribution related to environmental factors was also analyzed. I think that this paper is value to be published in Diversity. But two aspects should be noted and revised.
1. The authors mentioned that seven species (e.g. A. mieni, H. dumonti) appear to be endemic to Vietnam. It should not say “endemic” because there is no more extensive investigation in Southeast Asia.
2. In this paper, the authors analyzed the relationship of calanoid species to environmental variables. From the data in Table 2, the authors only list four variables. It is impossible to give an accurate analysis on this relationship. For calanoids, the nutrients, food and predation affect their distribution. The data in Table 2 may be problematic, especially pH and DO. The water bodies with low pH (3.39 and 2.64) should be acidic. As previous study, pH can deeply affect the calanoid distribution. There may be no calanoid distributon in so low pH. For DO, several water bodies were also very low (1.10 and 0.80), which inhibits the calanoid distribution. Meanwhile, the DO is impossible so high (15.10 and 12.70) in higher temperature. In Discussion, the authors only focused on conductivity rather than pH and DO. So I suggest that
a) the authors firstly verify whether the data of pH and DO are accurate;
b) if the authors could not determine the accuracy of the data, they should delete this part and keep the effect of conductivity;
c) if the authors can determine the accuracy of the data, the effect of pH and DO should be discussed, and give detailed data of each site in supplement including the calanoid distribution and environmental variables, especially low pH and DO;
d) if the authors have data on cyclopoids, I suggest that the authors added some data to discuss.
Author Response

(The authors gave the same response as above.)

Reviewer 3 Report
The paper is a nice example of an exhaustive field survey in a highly interesting and insufficiently known area in Vietnam; survey is exhaustive, carried out both in the rainy and in the dry season, and it updated the calanoid copepod checklist in Vietnam adding new and maybe still undescribed species. However there are some parts that must be improved; you will find all the suggestions in the attached revised manuscript in the comments to pdf. However, some general recommendation to improve the paper are as follows:
(1) line drawings are reported in methods but not used in the manuscript, while clustering is used but not reported in methods; pease check and be coherent
(2) the explanation of the use of CCA is quite confused and must be ameliorated; I suggest authors to check other papers on the argument (there are plenty) and be more technical and correct in the illustration of results; comments and suggestions are written in the commented pdf
(3) the discussion is quite detailed and narrative in the first part; maybe text can be shortened and more reference to the table will make reading more pleasant and less confused (in some parts I was lost among numbers...)
Main suggestion: it is not explained why, following author's results and opinion, the study area is so rich and interesting; more of the faunistical part of the discussion is dedicated to Vietnamese species richness, while I suppose it should be interesting to compare species richness of the stydy area in South Vietnam with the rest of the country and give suggestions to the determinants of its high diversity (maybe a habitat mosaic? is there any comparison with other river delta areas in Asia? or somewhere else?). Moreover a couple of sentence on the actul situation and protection status of the area, as well as the importance of calanoid species richness in the context of conservation and management of the area should be mentioned, increasing the interest of the manuscript. Are the habitats at risk? which is the level of anthropization? is it a natural protected area? As a reader, I'm curious to know such conservation details; as a copepodologist, I am astonished by the high species richness found, but no clear explanation is given on both.
Finally a recommendation on English style: please ask for a revision after correcting the manuscript. Better use 'sites' and not 'localities" throughout the text (their geographical meaning is different); please delete the articles in lists of habitat types; and so on.

Author Response

(The authors gave the same response as above.)

Reviewer 4 Report
I have some remarks as follows:
Methods: I did not see any line drawing in the manuscript (lines 88-89). Some technique of stacking was used for the photos? There is no explanation for the methods used for the scanning electron microscope (SEM). It seems that several SEMs were used.
The authors mention two possible new species in the results, but they do not explain why. If they conclude this point, it is necessary to say at least quickly why they consider that these two species do not belong to any one of the known species.
In the discussion part, the manuscript finishes abruptly, with no conclusions or final thoughts about the contribution that represents this manuscript. For example, the authors had an extensive sampling in the north of the country, but the south was not sampled. If they surveyed 87 localities, why did they find only a fraction of the species known for Vietnam? What will happen to the two species not identified, among many other points?
Table 2.- It will be interesting to know the number of measurements (N). Was 160X2 or 87X2?.
Fig. 5.- I do not understand; it is composed of two dendrograms, but there is no explanation, and lines 170-188 explaining this figure are also confusing. Moreover, the figure is not well balanced. It has ample blank space on the left side.
Conductivity is exceptionally variable and maybe should be worked apart. The deviation is more significant than average.
Line 110.- “applied to reveal differences in species numbers and diversity among two seasons. All statistical analyses were performed with SPSS 19.0 software. The significance level was 111 set as p < 0.05.”
Lines 153-160.- I do not understand. The authors said there was little difference between dry and rainy seasons. However, six species were found in the two seasons, and four species were found only in the dry season, whereas three were only during the rainy season.
In figure 4, I see only H. elegans similar in both seasons.
Line 197-198.- Maybe conductivity did not change in the same systems, but between systems, there is an enormous difference (11 to 3999 µS cm-1).
Lines 202-215.- I do not understand what the authors mean and from where they interpreted these four groups.
Lines 231-235.- Accordingly, with Fig. 6 and Table 4, the interpretation should be different. I see a positive relationship related to pH, which explains most of the Axis 1 of several species. A negative for other four. On the other side, conductivity, a related variable to Axis 2 and maybe the most important and variable environmental measurement has a positive correlation with seven species and a negative with five. It is not explained in the text.
Lines 344-346.- What about M. mekongensis and V. blanchei?
Author Response

(The authors gave the same response as above.)

Round 2
Reviewer 3 Report
I wish to thank the Authors who made an effort to ameliorate the quality of the paper. I suggest it can be accepted with minor revisions after Authors will take in account the suggestions (mainly grammar/style observations) listed below.
Table 1 - Please correct sites instead of localities inside Table1 column headers
row 112 - "were xplored" not "was explored"
row 117 "by cluster analysis" not "by the cluster.."
row 121 - "performed" not "done"
row 295 - "Jaccard" not "Jccard"
row 232 - "The first two axes of CCA ordination accounted for 85% of the variance..." please replace your sentence; round variance because below, row 236, 57.4+27.6 = 85 not 84.9...;)) then of course Tablle 4 gives the same problem, but you can let as it is being the program output
row 249 - "conductivity" not "conductivities"
rows 326-330 - "spiniform process" instead of "spinous process", I suggest
row 333 - "when we will have more specimens at disposition"
row 389 - "Unfortunately, due to the rapid..."
row 423 - "indicates that a total..."
row 430 - "China, with 17 species (51.5%) in common."
Reviewer 4 Report
In general terms the manuscript was really improved. However, I believe that more attention should be given to the conductivity. It has a strong variation in the systems, and I do not recommend presenting a mean of it (Table 2, and line 231). It can give a biased idea of it in the studied systems in Vietnam. Maybe for it should be better to leave only the variation and the strong increment of it in the dry season in all systems, not only the brackish and with marine influence (seems to be double value, in both low and high records).
Still some typos are present, for example, Line 205.- Should be “The Jaccard…..”
I believe Conclusions that are many, but are not mentioned here should be re-written again. I see the following: Lines 420-424.- It is a conclusion, but it repeats results. Lines 429-431 are a conclusion, but lines 431-435 again repeat results, and do not seem to be a conclusion.
After these corrections, I believe the manuscript could be accepted.
